# Long-Term Outcomes of T1 Colorectal Cancer after Endoscopic Resection

**DOI:** 10.3390/jcm9082451

**Published:** 2020-07-31

**Authors:** Eun Young Park, Dong Hoon Baek, Moon Won Lee, Gwang Ha Kim, Do Youn Park, Geun Am Song

**Affiliations:** 1Department of Internal Medicine, Pusan National University School of Medicine, Busan 49421, Korea; wyforever@hanmail.net (E.Y.P.); neofaceoff@hanmail.net (M.W.L.); doc0224@pusan.ac.kr (G.H.K.); gasong@pusan.ac.kr (G.A.S.); 2Biomedical Research Institute, Pusan National University Hospital, Busan 49421, Korea; 3Department of Pathology, Pusan National University School of Medicine, Busan 49421, Korea; pdy220@pusan.ac.kr

**Keywords:** colorectal cancer, endoscopic resection, overall survival, recurrence-free survival

## Abstract

Background and Aims: Endoscopic resection (ER) for submucosal invasive colorectal cancer (T1 CRC) can be grouped as curative ER (C-ER) and non-curative ER (NC-ER). Little is known about the long-term outcomes of patients in these two groups. Therefore, we have evaluated the long-term outcomes in endoscopically resected T1 CRC patients in C-ER and NC-ER groups. Methods: We conducted a retrospective study on 220 patients with T1 CRC treated with ER from January 2007 to December 2017. First, we investigated the long-term outcomes (5-year overall survival [OS] and recurrence-free survival [RFS]) in the C-ER group (*n* = 49). In the NC-ER group (*n* = 171), we compared long-term outcomes between patients who underwent additional surgical resection (ASR) (*n* = 117) and those who did not (surveillance-only, *n* = 54). Results: T1 CRC patients in the C-ER and NC-ER groups had a median follow-up of 44 (interquartile range 32–69) months. There was no risk of tumor recurrence and cancer-related deaths in patients with C-ER. In the NC-ER group, the 5-year OS rates were 75.3% and 92.6% in the surveillance-only and ASR subgroups, respectively. The hazard ratio (HR) for ASR in NC-ER vs. surveillance-only in NC-ER was statistically insignificant. However, RFS rates were significantly different between the ASR (97.2%) and surveillance-only (84.0%) subgroups. Multivariate analysis indicated a submucosal invasion depth (SID) of >2500 µm and margin positivity to be associated with recurrence. Conclusions: The surveillance-only approach can be considered as an alternative surgical option for T1 CRCs in selected patients undergoing NC-ER.

## 1. Introduction

Adoption of a nationwide screening program and recent advances in endoscopic instruments and techniques have led to the increased detection of early colon cancer (ECC) and reduction in colorectal cancer (CRC) incidence and mortality [1]. The diagnosis of submucosal invasive CRC (T1 CRC) is reported as 15–30% [2,3]. T1 CRC has a satisfactory prognosis, with a 5-year survival rate exceeding 90%; complete cure is achieved with endoscopic resection (ER) and/or radical surgery [4,5].

ECC is defined as the carcinoma confined to the mucosal or submucosal layer, regardless of the presence or absence of lymph node metastasis (LNM) [6]. The intramucosal CRC can be cured by a complete ER with safe and reliable en bloc resection, regardless of the tumor size and macroscopic type. However, 5–10% of patients with T1 CRC have LNM or distant metastasis; therefore, they require additional surgical resection with lymph node dissection (ASR) after ER to ensure complete tumor clearance [7,8]. According to the guidelines issued by the Japanese Society for Cancer of the Colon and Rectum (JSCCR) in 2016 [9], non-curative ER (NC-ER) for T1 CRC is defined based on the presence of at least one of the following criteria: (i) unfavorable histologic subtypes (poorly differentiated adenocarcinoma/mucinous carcinoma/signet ring cell carcinoma), (ii) deep submucosal invasion (submucosal invasion depth (SID) ≥1000 μm in non-pedunculated cancers), (iii) positive lymphovascular invasion (LVI) or (iv) positive or undetermined resection margins. In the absence of these factors, curative endoscopic resection (C-ER) is considered for T1 CRCs after ER [9]. 

It is unclear whether ASR or a surveillance-only approach after ER is an adequate treatment option for T1 CRC patients. The policy for managing pathologic T1 CRC after C-ER is surveillance-only; however, ASR is recommended in NC-ER cases due to the risk of LNM, according to the JSCCR guideline [9]. However, 90% of T1 CRCs do not involve LNM; therefore, implementing ASR in all patients with NC-ER may cause overtreatment [5]. Moreover, in actual clinical practice, some of these patients refuse to or cannot undergo surgery for various reasons, such as old age, several significant comorbidities and individual preference. Furthermore, ASR after ER is associated with an overall mortality of 1–5% and morbidity of 30%, especially in elderly people [10,11].

Although histopathologic features predicting the risk of LNM and residual cancer have been elucidated, the long-term outcomes in patients with T1 CRC undergoing ER and the characteristics of recurrence after ER remain unknown. We aimed to assess the long-term outcomes, viz. 5-year overall survival (OS) and recurrence-free survival (RFS) in patients with endoscopically resected T1 CRCs. Additionally, in the NC-ER group, we compared recurrence and the associated risk factors between the ASR and surveillance-only subgroups.

## 2. Methods

### 2.1. Patients

We conducted a retrospective study on 220 patients with T1 CRC treated with ER from January 2007 to December 2017. Following were the exclusion criteria: (a) surgical resection with LN dissection as the initial treatment (*n* = 129); (b) indeterminate tumor depth (*n* = 70); (c) pedunculated lesions (*n* = 30); (d) previous surgery owing to CRC (*n* = 5); (e) familial history of adenomatous polyposis (*n* = 3); (f) inflammatory bowel disease (*n* = 1); and (g) incomplete follow-up (*n* = 24). All patients, as of their last follow-up in December 2017, had an overall median follow-up period of 44 months (interquartile range [IQR] 32–69). All data related to patient background (age and sex), endoscopic features of the lesion (tumor size, location, macroscopic type, resection method and complications), histopathologic features (histologic type, SID, LVI and margin positivity) and follow-up status were obtained from the electronic data records. Data on cause and date of death were retrieved from the Korean Ministry of Statistics. This study was approved by the Institutional Review Board of the Pusan National University Hospital, Busan, South Korea (approval number: H-1902-025-076) and adhered to the Declaration of Helsinki.

### 2.2. Endoscopic Procedure

The ER procedure included endoscopic mucosal resection (EMR) and endoscopic submucosal dissection (ESD, including hybrid ESD). The EMR and ESD methods have been well-described previously [12]. All the surgical procedures were performed by experienced endoscopists (G.A.S and D.H.B) using the standard methods.

### 2.3. Histologic Assessment

All resected specimens were immediately stretched, pinned and fixed in 10% buffered formalin for 12–24 h and serially sectioned into 2-mm slices. All specimen slices were examined microscopically to evaluate resection margins and tumor characteristics, including tumor size, histologic type, SID LVI. Grade of differentiation was assessed on standard hematoxylin-eosin stained sections. SID was the distance from the lowest point of the muscularis mucosa (or ulceration surface) to the point of deepest tumor penetration (‘classic’ method) and the distance from the lowest point of the imaginary line of the plane of the muscularis mucosa to the point of deepest tumor penetration in cases of irregular (discontinuous or hypertrophic) or absent muscularis mucosae (‘alternative’ method) [9]. LVI was defined as the presence of clusters of malignant cells within an endothelium-lined vascular channel. “Positive lateral and vertical margin” were defined as exposure of the carcinoma at the submucosal margin of the resected specimen. Diagnosis was confirmed by a board-certified pathologist with expertise in gastrointestinal pathology (D.Y.P). The study pathologist reviewed all lesions.

### 2.4. Data Collection and Follow-up

Local recurrence following ER was defined as either any histologically identified colorectal neoplasia that occurred at the ER scar site or LNM detected using computed tomography (CT). Distant metastases were also detected using CT. Follow-up colonoscopies were performed annually after ER, as recommended in the JSCCR guidelines [9]. Physical examinations, blood tests and contrast-enhanced chest and abdominopelvic CT were recommended every 6 months in the first 3 years, and the patients were subsequently followed-up annually to evaluate the presence of LNM or distant metastasis. The start of the follow-up period was defined as the index date for ER, while the end was defined as either the date of death or 31 December 2017, whichever occurred first. Cancer-related deaths were defined as deaths due to CRC; cases involving death not related to cancer were examined to ascertain the cause. Patients were censored at the first occurrence of the outcome of interest, death or end of the study period, whichever came first.

### 2.5. Statistical Analyses

Descriptive statistics were presented as frequencies (%) for categorical variables and mean (±standard deviation) or median (IQR) for continuous variables. Continuous variables were compared using the Student’s t-test; categorical variables were compared using the Fisher’s exact test. *p* < 0.05 was considered statistically significant. Univariable and multivariable logistic regression analyses were performed to identify the patient- and tumor-related risk factors associated with tumor recurrence after initial ER.

OS and RFS were retrospectively assessed in each group of patients. RFS was defined as freedom from confirmed recurrence or death from the cancer, whereas OS was defined as freedom from death by any cause. To compare OS and RFS between groups, we constructed Cox regression models and Kaplan–Meier curves, and differences were compared using the log-rank test. We used Cox regression analysis to calculate hazard ratios (HRs) for death and recurrence for the following variables: risk stratification, age, sex, location, tumor size, configuration, resection method, SID, LVI and margin positivity.

All statistical analyses were performed by an independent statistician (Department of Biostatistics, Clinical Trial Center, Biomedical Research Institute, Pusan National University Hospital) using the R statistical package program (version 3.6.0; R Foundation for Statistical Computing, Vienna, Austria).

## 3. Results

### 3.1. Patient Characteristics and Clinicopathologic Features of the T1 CRC Treated with ER

The clinicopathologic features of all patients (*n* = 220) with endoscopically resected and histologically confirmed T1 CRC are summarized in Table 1. The cohort comprised 154 men and 66 women (median age, 65 [range 30–87] years).

There were 49 and 171 patients in the C-ER and NC-ER groups, respectively. The causes for undergoing NC-ER were SID > 1000 µm (*n* = 108, 63.2%); margin positivity (*n* = 54, 31.6%); and LVI (*n* = 11, 6.4%). There were no unfavorable histologic types since these lesions were initially surgically resected. The NC-ER group showed a higher proportion of rectal cancers (20/171, 11.7%) and larger tumor size (15.1 ± 7.1 mm) than the C-ER group, although the difference was insignificant (*p* = 0.175 and *p* = 0.085, respectively). However, lesions in the NC-ER group showed a greater SID (2261.3 ± 1270.9 vs. 599.7 ± 273.4 µm) and higher proportion of LVI (11/171, 6.4% vs. 0/49, 0%) than those in the C-ER group (*p* = 0.001 and *p* = 0.009, respectively). All lesions with positive margins were included in the NC-ER group, and the difference was significant (31.6% vs. 0%, *p* < 0.001). There were no significant differences between the groups with respect to age, sex, macroscopic type, resection method and procedure-related adverse events. Tumor recurrence (11/171, 6.4%, *p* = 0.129) and one cancer-related death was observed only in the NC-ER group.

### 3.2. Long-Term Outcome of Patients in the C-ER and NC-ER Groups

To evaluate long-term outcomes, patients in the C-ER and NC-ER groups were further divided into the ASR and surveillance-only subgroups (Figure 1).

(1)Long-term outcomes in the C-ER group:

No tumor recurrence or cancer-related death was observed. Twelve of 49 (24.5%) patients showed concern about LNM and opted for surgery; however, LNM or recurrence was not observed in the ASR subgroup during follow-up (median, 44 [IQR: 32–69] months).

(2)Long-term outcomes in the NC-ER group

In this group, 117 (68.4%) patients underwent ASR, and 54 (31.6%) opted for surveillance-only during follow-up. ASR was performed in 115 patients (98.3%) with an SID ≥ 1000 μm, 11 patients (9.4%) with LVI and 45 patients (38.5%) with margin positivity. The long-term outcomes in the NC-ER group based on treatment strategy are shown in Figure 2. In Figure 2a, the Kaplan–Meier curves of OS showed no significant difference between the ASR and surveillance-only subgroups, with an HR of 2.057 (95% confidence interval [CI] 0.689–6.136; *p* = 0.19) for the surveillance-only vs. ASR subgroup. The 5-year OS rates were 75.3% (95% CI 57.8–98.1) and 92.6% (95% CI 86.3–99.2) in the surveillance-only and ASR subgroups, respectively. The Kaplan–Meier curves for RFS are presented in Figure 2b. The ASR subgroup showed better RFS than the surveillance-only subgroup, given the HR of 6.127 (95% CI 1.623–12.134; *p* = 0.0023) for the surveillance-only vs. ASR subgroup. The 5-year RFS rates were 84.0% (95% CI 72.4–97.5) and 97.2% (95% CI 94.3–100) in the surveillance-only and ASR subgroups, respectively.

Additionally, we compared clinicopathologic characteristics between lesions with and without recurrence (Table 2) and analyzed the risk factors for recurrence in the surveillance-only subgroup (Table 3). Eight of 54 patients (14.8%) showed recurrence. In the multivariate analysis, an SID > 2500 µm (HR, 7.298; CI, 1.253–42.500; *p* = 0.027) and margin positivity (HR, 7.189; CI, 1.033–50.029; *p* = 0.046) were significantly associated with recurrence. In the ASR subgroup, no variable was found to be a significant risk factor for recurrence in the univariate and multivariate analyses (see Appendix A).

### 3.3. Characteristics of Recurrent T1 CRC

The details of patients with cancer recurrence are summarized in Table 4. The median time to recurrence was 24.8 (range 4.1–82.2) months. While patients in the C-ER group did not show recurrence, all patients with recurrence (*n* = 11) belonged to the NC-ER group. Local recurrence was observed in eight patients and distant metastasis in three (two with liver metastasis and one with lung and bone metastases). Eight patients belonged to the surveillance-only subgroup (8/54, 14.8%) and three patients to the ASR subgroup (3/117, 2.7%).

In the surveillance-only subgroup, one patient had liver metastasis, which was detected 82.2 months after the index ER. The other seven cases involved local recurrence, observed after a median of 24.8 (IQR 4.1–81.9) months. Of these, the earliest local recurrence was detected 4.1 months after the index ER, which was located in the rectum, with an SID > 2500 µm and pathologically positive vertical margins. Two recurrence cases were observed after 5 years (one, local recurrence; one, distant metastasis). The remaining six recurrence cases involved local recurrence, detected within 3 years. In the ASR subgroup, all three recurrences (two, distant metastases; one, local recurrence) were observed within 1 year of the index ER. Distant metastases involved liver metastasis and lung and bone metastases. Local recurrence was observed at the anastomosis site of ASR.

In total, 14 patients (14/220, 6.4%) died during the follow-up period, and all death cases belonged to the NC-ER group. There were only three deaths among patients with CRC recurrence, and cancer-related death was confirmed in only one patient with bone metastasis, while two patients died due to cerebrovascular disease and pneumonia. In patients without recurrence (*n* = 11), patients died of cerebrovascular disease (*n* = 2), accidental reasons (*n* = 2), dementia (*n* = 2), pneumonia (*n* = 2), intestinal infection (*n* = 1), hypopharyngeal cancer (*n* = 1) or lung cancer (*n* = 1).

## 4. Discussion

The present study revealed two main results for long-term outcomes of T1 CRCs after ER. First, we investigated the long-term outcomes in C-ER group. There was no risk of tumor recurrence and cancer-related deaths in patients with C-ER. Second, we compared long-term prognosis in the NC-ER group of T1 CRC patients, with or without ASR. The difference in OS between ASR and surveillance-only subgroups was statistically insignificant. However, RFS rates were significantly different between the ASR (97.2%) and surveillance-only (84.0%) subgroups. Multivariate analysis indicated a submucosal invasion depth (SID) of >2500 µm and margin positivity to be associated with recurrence. These results suggest surveillance-only approach can be considered as an alternative surgical option for T1 CRCs in selected patients undergoing NC-ER.

T1 CRC has shown steady increase in the incidence rate and accounts for 17% of the total CRC cases. Moreover, T1 CRC are confirmed in about 0.6% of the cases after ER [13]. However, the long-term outcomes in patients with T1 CRC after ER, such as the OS and RFS, and characteristics and types of tumor recurrence, remain unknown. Several studies have reported histopathologic and prognostic factors for predicting LNM in T1 CRC patients treated with surgical resection and lymph node dissection [14,15]. Despite the high clinical and practical relevance, there are only few reports on the follow-up data after ER for patients with T1 CRC [4,5,16].

In this retrospective cohort study, we have examined the long-term outcomes in patients with T1 CRC after ER. Patients in the C-ER group, fulfilling histopathologic factors described in the JSCCR guidelines, showed excellent long-term outcomes with ER alone, did not show tumor recurrence, and none of the patients who underwent ASR showed LNM, consistent with previous reports [4,17]. Therefore, our analysis supports that T1 CRC, which satisfies the curative criteria of the JSCCR guidelines, shows no increased risk of recurrence without ASR. However, this treatment strategy can be allowed after comprehensive evaluation of histological diagnosis by expert gastrointestinal pathologists to ensure surveillance-only. Yoda et al. reported that after reexamination of original pathologic specimen, LVI was detected in 0.8% (1/126) of the cases who were earlier classified as C-ER [5].

The investigation of long-term prognosis in the NC-ER group of T1 CRC patients, with or without ASR, is the highlight of our study. The difference in OS between ASR and surveillance-only subgroups was statistically insignificant. A few studies enrolling large number of patients and long follow-up periods have reported similar results as found in our study. The review of patients with T1 CRC in the SEER database (The Surveillance, Epidemiology and End Results, the National Cancer Institute, USA) showed similar risk of death in the ASR and surveillance-only groups, consistent with our results, after accounting for age and comorbidities and adjusting for propensity quintile [18]. In the report by Yamashita et al. difference in the 5-year OS rates in the ASR and surveillance-only subgroups in patients with NC-ER was statistically insignificant [19]. Surveillance-only and close follow-up after ER in the NC-ER group may serve as good alternative treatment options rather than surgery, especially in patients with old age or significant comorbidities. Various circumstances should be considered to determine the best approach for patients diagnosed with T1 CRC, and therefore, further studies should be ascertained in large cohort studies with long-term follow-up periods to confirm the benefits.

Further, our analysis provides data on the recurrence of patients in the NC-ER group. Of the 117 patients in the NC-ER group, RFS was significantly lower in the ASR subgroup than in surveillance-only subgroup. Similar to our results, recurrence after ER of T1 CRC patients was 9.5% in high-risk lesions (poor differentiation, SID > 1000 µm, LVI and positive resection margin), which was higher than that in patients with low risk lesions (1.2%) in a recent meta-analysis [20]. This suggests that ASR should be warranted in the T1 CRC patients undergoing NC-ER for the aspect of recurrence. However, the beneficial rate (residual tumor and LNM) in patients with surveillance-only vs. risk rate (post-operative morbidity and mortality) in patients with ASR should be balanced, because these two factors are weighed against each other in the real-world clinical practice. Benizri et al. [10]. reported the rates of benefit and risk of ASR after ER to be 10.9% and 25%, respectively. For the beneficial rate by performing subsequent ASR, Choi et al. [21]. Reported that 14.3% (24/168) of the T1 CRC patients in NC-ER group to have benefited and Rickert et al. [22]. Showed beneficial rate of 41% for residual tumor and 8.6% for LNM. Whereas, for the aspect of risk rate, several studies have reported the postoperative complication rate of 18.8–31.8% in ASR after ER [10,11,22]. The surgical mortality rates of 1.9–6.5% for colon cancer, 3.2–9.8% upon total mesorectal resection for rectal cancer [23,24] and 0.8% upon receiving ASR after ER have been observed [25]. Furthermore, ASR may be unnecessary in most patients because of the overall 10% rate of LNM in T1 CRCs [5]. Therefore, various circumstances need to be considered to determine the best approach for patients diagnosed with T1 CRC, and comprehensive treatment decision should be made based on other factors, such as age, significant comorbidities and physical activity levels in medically fit patients.

Our study elaborates recurrence in ASR and surveillance-only subgroup of NC-ER. Recurrence was observed in 14.8% (8/54) of the patients in the surveillance-only group. In the multivariate analysis with logistic regression, the SID ≥ 2500 µm and margin positivity were found to be independent risk factors for recurrence. According to the JSCCR guidelines, SID ≥ 1000 µm indicates ASR, and several studies have identified deep SID as the most frequent indication for ASR after ER [26,27]. The deeper the SID, the higher the risk of recurrence and risk of recurrence based on SID > 1000 µm showed increased incidence of metastasis with the relative risk of 3.0–5.93 in previous meta-analysis [28,29]. However, recent studies have shown that even in T1 CRC with SID ≥ 1000 µm, the rate of LNM was only about 1–2% in the absence of other risk factors [30,31]. Consistent with those results, our study also showed cutoff value of SID > 2500 µm, not SID > 1000 µm, to be associated with risk of recurrence. Thus, this result indicates that surveillance-only strategy after ER without other risk factors, except for only deep submucosal invasion colorectal cancer (pT1b), may serve as a potential treatment option. However, lesions with SID ≥ 2500 µm was associated with recurrence and further studies are warranted to confirm the exact depth related to recurrence. Margin positivity is the major indication for ASR after ER in T1 CRC because it is significantly associated with residual disease and local recurrence due to the possibility of regrowth of remaining tumor cells [32]. Previous studies have demonstrated that positive margin status, especially vertical margin, is significantly associated with residual disease in patients with endoscopically resected T1 CRC [33,34]. In this study, we also demonstrated that margin positivity significantly associates with recurrence, and therefore, ASR should be recommended in margin positive T1 cancer.

This study has some limitations. First, the outcome of the study was limited by its retrospective nature, along with selection bias. However, it would not be ethical to conduct a randomized study to compare the long-term outcomes in surveillance-only vs. ASR in patients with NC-ER. Second, the statistical power would be insufficient due to the small number of death and recurrence. Therefore, a study based on the population-oriented large cohort would represent the best way to assess the long-term outcomes in current practice. Third, the histological differentiation status and tumor budding were not considered as risk factors of OS and RFS in NC-ER group. We could not analyze tumor differentiation because patients with poor histologic type tumors underwent surgical resection with lymph node dissection as the first-line treatment. Additionally, we could not analyze the effect of tumor budding on OS and recurrence due to unavailability of pathology for about 50% of the patients in the cohort. However, tumor budding neither allows prediction of the adverse prognostic events [35], nor serves as a reliable indicator in regular clinical practice. Therefore, additional data/analysis are required to determine whether these criteria are independent prognostic factors for reliable utility in the pathologic laboratories.

In conclusion, T1 CRC included in the C-ER group according to the JSCCR guidelines has no increased risk of recurrence. While OS of patients in the NC-ER group was not affected by ASR, RFS was significantly lower in the ASR subgroup than that in the surveillance-only subgroup of the NC-ER group. SID ≥ 2500 µm and margin positivity were identified as independent risk factors for prediction of recurrence. Surveillance-only in the NC-ER group may be considered as an alternative surgical option for T1 CRCs in selected NC-ER patients.

## Figures and Tables

**Figure 1 jcm-09-02451-f001:**
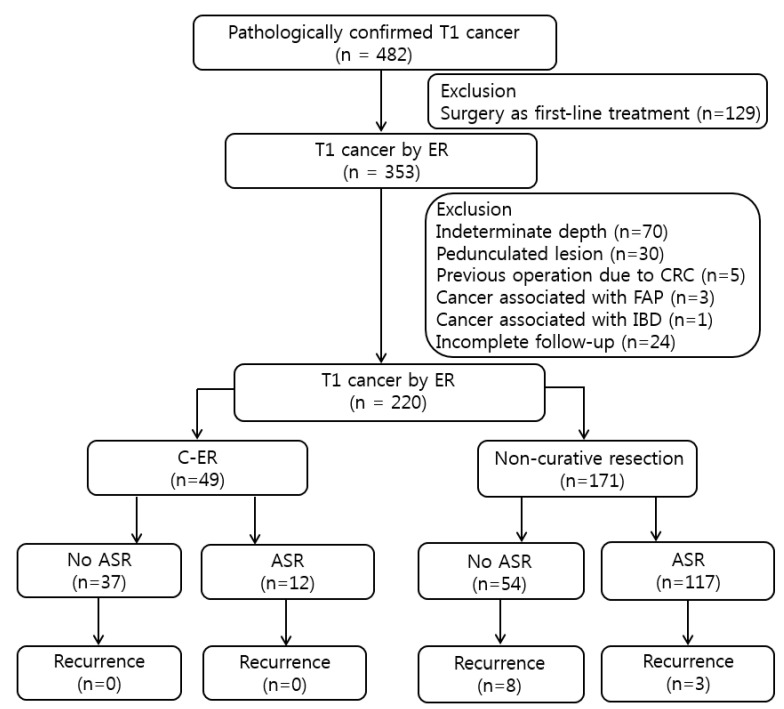
Clinicopathologic features of total 220 submucosal invasive colorectal cancers (T1 CRCs). FAP—familiar adenomatous polyposis; IBD—inflammatory bowel disease; C-ER—curative-endoscopic resection; NC-ER—non-curative endoscopic resection; ASR—additional surgical resection.

**Figure 2 jcm-09-02451-f002:**
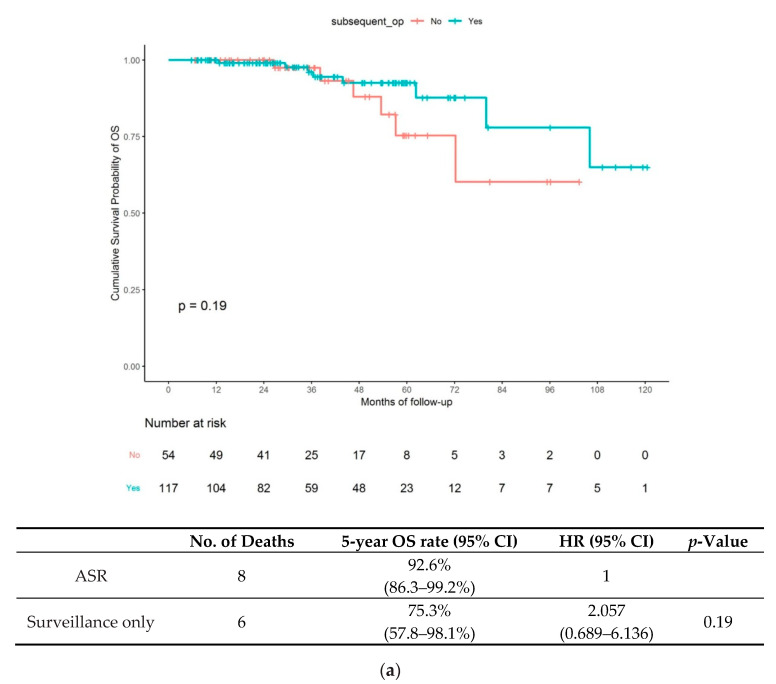
(**a**) Hazard ratios and 95% confidence intervals of overall survival for additional curative surgery vs. surveillance-only; (**b**) hazard ratios and 95% confidence intervals of recurrence free survival for additional curative surgery vs. surveillance-only. Overall, survival and recurrence free survival according to treatment strategy for the NC-ER group according to NC-ER followed by additional surgical resection (ASR) and NC-ER with surveillance-only; (**a**) overall survival (upper) and (**b**) recurrence-free survival (lower).

**Table 1 jcm-09-02451-t001:** Clinicopathologic features of the 220 submucosal invasive colorectal cancers.

Variable	C-ER, *N* (%) (Overall *N* = 49)	NC-ER, *N* (%) (Overall, *N* = 171)	*p*-Value
Age (years), mean ± SD	64.5 ± 8.8	63.9 ± 10.0	0.688
Sex, *n* (%)			>0.999
Male	34 (69.4)	120 (70.2)	
Female	15 (30.6)	51 (29.8)	
Location, *n* (%)			0.175
Colon	47 (95.9)	151 (88.3)	
Rectum	2 (4.1)	20 (11.7)	
Size, mm, mean ± SD	13.2 ± 6.4	15.1 ± 7.1	0.085
Macroscopic type, *n* (%)			0.190
Sessile	17 (34.7)	87 (50.9)	
Flat	9 (18.4)	24 (14.0)	
LST-G	6 (12.2)	12 (7.0)	
LST-NG	17 (34.7)	48 (28.1)	
Resection method, *n* (%)			0.195
ESD	39 (79.6)	152 (88.9)	
EMR	10 (20.4)	19 (11.1)	
Adverse events			0.444
Acute bleeding, *n* (%)	1 (2.1)	11 (6.6)	
Delayed bleeding, *n* (%)	0 (0.0)	2 (1.2)	
Perforation, *n* (%)	1 (2.0)	3 (1.8)	
Pathology			
Well or moderately differentiated	49 (100)	171 (100)	
Poorly differentiated/mucinous/signet ring cell carcinoma	0 (0.0)	0 (0.0)	
Submucosal invasion depth	599.7 ± 273.4	2261.3 ± 1270.9	0.001
Lymphovascular invasion	0 (0)	11 (6.4)	0.009
Margin positivity	0 (0)	54 (31.6)	<0.001
Recurrence	0 (0)	11 (6.4)	0.129
Death	0 (0)	14 (8.2)	0.043
Cancer-related death	0 (0)	1 (0.6)	>0.999

C-ER—curative endoscopic resection; EMR—endoscopic mucosal resection; ESD—endoscopic submucosal dissection; LST-G—laterally spreading tumor–granular type; LST-NG—laterally spreading tumor–non-granular type; NC-ER—non-curative endoscopic resection; SD—standard deviation.

**Table 2 jcm-09-02451-t002:** Comparison of clinicopathologic characteristics in the surveillance-only subgroup of the non-curative endoscopic resection (NC-ER) group according to recurrence.

	Recurrence *N* (%) (Overall *N* = 8)	No recurrence *N* (%) (Overall *N* =46)	*p*-Value
Age (years), mean ± SD	72.5 ± 8.9	65.2 ± 11.0	0.062
Sex, *n* (%)			>0.999
Male	6 (75.0)	35 (76.1)	
Female	2 (25.0)	11 (23.9)	
Location, *n* (%)			0.577
Colon	8 (100)	39 (84.8)	
Rectum	0 (0)	7 (15.2)	
Size (mm), mean ± SD	19.3 ± 9.3	15.0 ± 7.4	0.253
Macroscopic type, *n* (%)			0.896
Sessile	4 (50.0)	17 (37.0)	
Flat	1 (12.5)	8 (17.4)	
LST-G	0 (0)	6 (13.0)	
LST-NG	3 (37.5)	15 (32.6)	
Resection method, *n* (%)			0.588
ESD	6 (75.0)	40 (87.0)	
EMR	2 (25.0)	6 (13.0)	
Pathology			
Well and moderately differentiated	8 (100)	46 (100)	
Submucosal invasion depth	1977 ± 1443	1481 ± 770	0.372
Lymphovascular invasion			>0.999
Positive	0 (0)	5 (10.6)	
Negative	8 (100)	41 (89.4)	
Margin			0.275
Positive	2 (28.6)	7 (14.9)	
Negative	6 (71.4)	39 (85.1)	

EMR—endoscopic mucosal resection; ESD; endoscopic submucosal dissection; LST-G—laterally spreading tumor–granular type; LST-NG—laterally spreading tumor–non-granular type; SD—standard deviation; NC-ER—non-curative endoscopic resection.

**Table 3 jcm-09-02451-t003:** Risk factors for recurrence in the surveillance-only subgroup of the NC-ER group.

	No. of Patients	No. of Events	Univariate			Multivariate		
			OR	95% CI	*p*-Value	OR	95% CI	*p*-Value
Age (years)								
≥65	36	7	3.328	0.386–28.726	0.245			
<65	18	1	1					
Sex								
Male	41	6	1					
Female	13	2	1.279	0.232–7.037	0.777			
Size (mm)								
≥15	28	5	0.832	0.167–4.139				
<15	26	3	1		0.822			
Submucosal invasion depth								
≥2500	10	3	5.383	1.079–26.858	0.040	7.298	1.253–42.500	0.027
<2500	44	5						
Lymphovascular invasion					0.999			
Positive	5	0	1					
Negative	49	8	0	0.000–Inf				
Margin					0.055			0.046
Positive	7	2	5.390	0.965–30.096		7.189	1.033–50.029	
Negative	47	6	1			1		

CI—confidence interval; OR—odds ratio; NC-ER—non-curative endoscopic resection.

**Table 4 jcm-09-02451-t004:** Details of the 11 patients with recurrence after endoscopic resection.

Patient	Sex/Age (years)	Location	Size, mm	Resection Method	Resection Type	Initial Histology	Submucosal Invasion Depth	Lympho-Vascular Invasion	Margin Status at ER	Operation	Time to Recurrence, Month	Recurrence Type	Recurrence Treatment	Death	Death Cause
1	Female/73	RC	25	ESD	En bloc	Mode-diff	1450	–	–	Yes	9.1	Distant meta(liver meta)	Chemotherapy	Yes	Cerebrovascular disease
2	Male/72	Rectum	35	ESD	Piecemeal	Well-diff	2750	–	VM	No	4.1	Local recurrence	Op. refuse	Yes	Pneumonia
3	Male/65	LC	15	ESD	En bloc	Mode-diff	3000	–	VM	Yes	9.1	Distant meta(lung and bone meta)	Chemotherapy	Yes	Colon cancer
4	Male/71	RC	30	ESD	Piecemeal	Well-diff	440	–	VM	No	25.9	Local recurrence	Endoscopic resection	No	
5	Male/66	LC	25	EMR	Piecemeal	Mode-diff	1000	–	VM	Yes	10.8	Local recurrence	Operation	No	
6	Female/58	LC	25	ESD	En bloc	Well-diff	1250	–	–	No	37.8	Local recurrence	Operation	No	
7	Male/66	LC	25	ESD	En bloc	Mode-diff	1475	–	–	No	82.2	Distant meta(liver meta)	Chemotherapy	No	
8	Male/75	LC	20	EMR	En bloc	Mode-diff	2250	–	–	No	81.9	Local recurrence	Op. refuse	No	
9	Female/81	LC	14	EMR	En bloc	Mode-diff	2500	–	–	No	24.8	Local recurrence	Operation	No	
10	Male/87	LC	15	ESD	En bloc	Mode-diff	4150	–	–	No	12.0	Local recurrence	Operation	No	
11	Male/70	LC	10	EMR	En bloc	Mode-diff	1000	–	–	No	15.0	Local recurrence	Operation	No	

EMR—endoscopic mucosal resection; ESD—endoscopic submucosal dissection; LC—left colon; RC—right colon; VM—vertical margin positivity; meta—metastasis; Well-diff—well-differentiated; Mode-diff—moderately differentiated.

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
