# Peer review of "Long-Term Outcomes of T1 Colorectal Cancer after Endoscopic Resection"

_jcm, 2020, doi:10.3390/jcm9082451_

Round 1
Reviewer 1 Report
This is a well written paper that explores the clinical outcomes of patients undergoing endoscopic resection for colorectal carcinoma. The clinically predictive features include depth of invasion >2500 um and margin positivity, which are associated with an increased risk of recurrence. Given the significance of this finding, it is important to have a complete explanation of how the depth of invasion is measured. Measuring the depth of invasion can be challenging given the varying presentations of colorectal carcinom (exophytic, flat, depressed). Is the invasion measured from the top of the mucosa, the bottom extent of the muscularis mucosae, or something else? Given that the main features in this paper are related to tumor histology, it seems disappointing that the method of measurement is not described precisely by the study pathologist. In order for this study to be replicated and applied broadly in clinical practice, this should be described precisely. The authors also suggest that histologic features such as tumor budding do not predict adverse prognostic events, when several studies have demonstrated the importance of tumor budding in predicting lymph node metastases - which would argue for resection for curative intent. It should be acknowledged, at minimum, that there is literature to support the importance of tumor budding. The pathologist on the study should also be able to count and enumerate tumor buds for the cases in question. Overall, this is a very important study that brings to light important objective histologic measures that impact patient outcome, and will be important to determining how these cases are evaluated and reported.
Author Response
Thank you for your kind comments and thoughtful questions.
The following sentence has been inserted on page 2 [Line number: 91], detailing measurement of the depth of submucosal invasion:
SID was the distance from the lowest point of the muscularis mucosa (or ulceration surface) to the point of deepest tumor penetration (‘classic’ method), and the distance from the lowest point of the imaginary line of the plane of the muscularis mucosa to the point of deepest tumor penetration in cases of irregular (discontinuous or hypertrophic) or absent muscularis mucosae (‘alternative’ method).
Guidelines recommending further surgical procedures after endoscopic resection have been established, using a combination of risk factors for LN metastasis (such as unfavorable histologic subtypes, SID, LVI, positive or undetermined resection margins, and tumor budding) in patients with T1 CRC. However, these guidelines remain controversial, specifically the status of tumor budding. We believe that it is important to evaluate risk factors and their significance in patients with T1 CRC, particularly tumor budding. Therefore, we are collecting ESD specimens of T1 CRC. In the near future, we plan to publish our data whether tumor budding is independent prognostic factors for reliable utility in the pathological laboratories.
Reviewer 2 Report
Research investigation is minimal on this topic. It is further advancement from research conducted by Asayama et al., 2016. Investigation conducted by park et al is going to further strengthen the research in the area of colorectal cancer.
Authors have addressed the good question. Genuinely very little is known about the long term outcomes of patients in T1 colorectal cancer.
Line number: 21 howeverm should be replace with however.
Authors have elucidated long term outcomes of T1 colorectal cancer after endoscopic resection. Manuscript is really good, very well written. However, i have one minor suggestion, authors should pay attention to english.
Author Response
Thank you for your kind comments.
We have corrected “Line number: 21 howeverm” as “however”.
I have my manuscript checked by a professional English editing service. A certificate of editing is attached as a file.

Reviewer 3 Report
Dr. Park and colleagues performed a retrospective study to assess the long-term outcomes of patients with endoscopically resected submucosal invasive colorectal cancer. They found that submucosal invasion depth ≥ 2500μm and margin positivity were independent risk factors for prediction of recurrence, and concluded that surveillance-only in the non-curative-ER group may be considered as an alternative surgical option in selected non-curative-ER patients.
The study is well conducted and well written. Statistical analysis was accurate and clearly presented.
I have some minor comments:
- methods: I'd start this paragraph with a sentence like "We conducted a retrospective study on 220 patients with T1 CRC treated with ER from 18 January 2007 to December 2017. "
- table 1, last row: p value should be indicated for uniformity
- some typos and sentences should be changed, i.e. the first sentence of the abstract ("Endoscopic resections for T1 CRC can be grouped as...")
Author Response
Thank you for your kind comments.
In the method section, the following sentence was deleted: "Eventually, 220 T1 CRC patients who underwent C-ER were included." And The following sentence has been added to the limitation section on page 2 [line number 68]: "We conducted a retrospective study on 220 patients with T1 CRC treated with ER from January 2007 to December 2017."
We have corrected “table 1, last row: p value = n-s” as follow:“p value >0.999”
We have corrected the first sentence of the abstract [line number 14] as follow: “Endoscopic resection (ER) for submucosal invasive colorectal cancer (T1 CRC) can be grouped as: curative ER (C-ER) and non-curative ER (NC-ER).”